# Anomalous Ca Content Dependence of Dielectric Properties of Charge-Ordered $Pr_{1−x}Ca_xMnO_3$ as a Signature of Charge-Ordered Phase Modulation

Ankit Kumar Singh and Partha Sarathi Mondal *

Department of Applied Sciences and Humanities, National Institute of Advanced Manufacturing Technology, Ranchi 834003, India
* Correspondence: psmondalmbox@gmail.com

**Abstract:** Low-temperature dielectric properties of charge/orbital-ordered manganite, $Pr_{1−x}Ca_xMnO_3$ for $0.40 \leq x \leq 0.50$, was investigated systematically as a function of Ca content, $x$. The Ca content dependence of dielectric permittivity and dissipation factor exhibited distinct maxima near $x \sim 0.45$. The overall dielectric response of charge-ordered $Pr_{1−x}Ca_xMnO_3$ was dominated by dielectric polarization induced by polaron hopping and exhibited thermally activated relaxation behaviour. The thermally activated dielectric relaxation behaviour over the investigated temperature range was further analysed with the help of two models: the small polaron hopping model and the Mott three-dimensional variable range hopping model. The estimated polaron transport parameters also displayed non-monotonic variation with $x$ and exhibited a broad minima between $x = 0.425$ and 0.45. Considering the previous work reported so far, the charge order pattern of $Pr_{1−x}Ca_xMnO_3$ below $x = 0.425$ was most likely to be of Zener-polaron type, while near $x = 0.50$ was checker-board type and for in-between compositions; neither pure checker-board type nor pure Zener-polaron type can be considered a ground state. The observed results suggest that a modulation of the checkerboard-type charge/orbital ordering pattern in $Pr_{1−x}Ca_xMnO_3$ possibly takes place in the Ca content range of investigation, $0.40 \leq x \leq 0.50$.

**Keywords:** manganite; charge/orbital ordering; dielectric properties; polaron



## 1. Introduction

Mixed valence perovskite manganite has renewed considerable interest after the discovery of colossal magnetoresistance [1,2]. The physical property of manganite involves a complex interplay among the charge, spin, and orbital degree of freedoms of an electron [1]. It leads to the appearance of a wide variety of competitive electronic phases with comparable energy scale across a phase diagram, principally, the antiferromagnetic insulating (AFMI) phase, ferromagnetic metallic (FMM) phase, ferromagnetic insulating (FMI) phase, paramagnetic insulating (PMI) phase, etc. The delicate balance among these diverse phases can be easily tuned through minute external or internal perturbations, and accordingly, physical properties can be manipulated [1–4]. Of particular importance to the present investigation is the charge/orbital ordered insulating (COI) phase in $Pr_{1−x}Ca_xMnO_3$ (PCMO), which appears along with the CE-type of antiferromagnetic (AF) order in almost all perovskite-type manganite near the half doping ($x = 0.5$), where the presence of an equal amount of $Mn^{3+}$ and $Mn^{4+}$ is expected [5,6]. In general, for large bandwidth perovskite manganite, the COI phase appears only in a narrower doping range near $x = 0.5$ [6]. However, for a narrow bandwidth manganite, such as PCMO, it is quite striking that the COI phase exists over a wider range of doping, i.e., $0.3 \leq x \leq 0.75$ [7–9]. The high-temperature phase of PCMO is PMI, where all the Mn sites appear to be equivalent. Below a critical temperature (e.g., $T_{CO} \sim 250$ K for $x = 0.5$), the charge/orbital ordering transition occurs, being concomitant with a structural transition and two distinct charge sites, $Mn^{3+}$ and

$Mn^{4+}$, which are stabilised by strong Coulomb interaction along with the co-operative Jahn–Teller distortion of $MnO_6$ octahedra [9–12]. Further lowering of the temperature of the PCMO also leads to exhibition of a paramagnetic to antiferromagnetic transition (e.g., $T_N \sim 175$ K for $x = 0.5$), which is also associated with an incommensurate to commensurate charge ordering transition [12].

The ground state of PCMO near half filling is still an unresolved question, despite the existence of extensive studies. Goodenough [5] proposed checkerboard-type (CB-type) charge/orbital ordering, which seems to realise exactly at half filling, $x = 0.5$. This phase assumes a Mn-site-centred charge/orbital order. Another state, namely, bond-centred charge order, was proposed by Daoud-Aladine et al. from the analysis of neutron diffraction data close to $x = 0.40$ [13]. In between, neither pure CB-type nor pure Zener polaron can be considered as a ground state; instead, the coexistence of the CB-type of charge/orbital and Zener-polaron-type (ZP-type) orderings can be present [14–16]. Interestingly, the coexistence of CB-type (site-centred) and ZP-type (bond-centred) ordering in intermediate compositions $0.40 < x < 0.50$ can break the inversion symmetry and can give rise to ferroelectric instability [16]. Several experimental efforts have been devoted in order to investigate the role of charge/orbital ordering in intrinsic dielectric response of PCMO, but ferroelectric instability due to charge/orbital ordering in PCMO has never been found experimentally [17–21]. The anomalous increase in dielectric constant at the vicinity of $T_{CO}$ in PCMO affirms the significant role of charge/orbital ordering in intrinsic dielectric response [19,20]. Lopes et al., by measuring the electric field gradient (EFG) in PCMO, revealed a typical signature of phase transition involving long range ordering of local dipoles, confirming ferroelectric instability at least in the local scale due to charge/orbital ordering [22]. Therefore, it is expected that the nature charge/orbital ordered phase in PCMO is not essentially the same across the doping range $0.40 \leq x \leq 0.50$; instead, there is a possibility of evolution/modulation of the charge ordering pattern. In spite of such expectations, there is no one reported clear piece of evidence for such modulation of the charge ordering pattern. However, in our earlier report, we showed abrupt changes in thermoelectric power and saturation magnetisation in PCMO at $x \sim 0.425$, which is interpreted as a phase boundary between two different charge/orbital ordered phases [23].

In the present study, we considered Ca content dependence of the crystal structure and dielectric properties in the charge-ordered manganite PCMO near half filling ($0.40 \leq x \leq 0.50$) with a close variation of Ca content, $x$. Abrupt change in the dielectric properties near $x \sim 0.45$ was demonstrated. The distinct maxima in dielectric permittivity and in dissipation factor were observed near $x \sim 0.45$. The observed dielectric relaxation in PCMO originated from thermally activated polaron hopping, and transport parameters associated with polaron hopping also exhibited an abrupt change below $x \sim 0.45$. The observed results were interpreted as a signature of phase transition between two different charge/orbital ordered phases, possibly between CB-type charge/orbital ordered phase and ZP-type ordered phase.

## 2. Results

The systematic investigation of the crystal structure of PCMO for $0.4 \leq x \leq 0.5$ was conducted by employing high-resolution X-ray diffraction (HRXRD). All the compositions were found to be single phased, and at room temperature, they exhibited an orthorhombic unit cell with a *Pnma* space group. Estimated unit cell parameters, *a*, *b*, and *c* and cell volume *V* from the refinement of HRXRD data were found to be decreased monotonically as Ca content, $x$, gradually increased (Figure 1), which was in accordance with the smaller ionic radius of $Ca^+$ than that of $Pr^+$.

The real part of dielectric permittivity, $\epsilon'(\omega, T)$, versus temperature ($T$) patterns measured for several frequencies across the temperature range $4-150$ K are shown in the upper panels of Figure 2 for representative samples of PCMO with Ca contents: $x = 0.40$, 0.45, and 0.5. The dissipation factor $D(\omega, T)$ versus the $T$ plot for those compositions are also shown in the lower panels of Figure 2. A typical feature of dielectric relaxation behaviour for all the samples was observed clearly. At temperatures above 100 K, $\epsilon'(\omega, T)$ reached a

value as high as $\varepsilon'{\sim}10^4$ with almost no variation with temperature and frequency, and with decreasing temperature, $\epsilon'(\omega, T)$ showed step-like decrease and attained a temperature- and frequency-independent value of about $\epsilon'{\sim}30-50$. The characteristic temperature ($T$) at which the inflection in $\epsilon'(\omega, T)$ occurred was also associated with a peak in the corresponding $D(\omega, T)$ versus the $T$ plot, which gradually shifted towards the higher temperature with increasing frequency. This confirmed the thermally activated dielectric relaxation process [17,18].

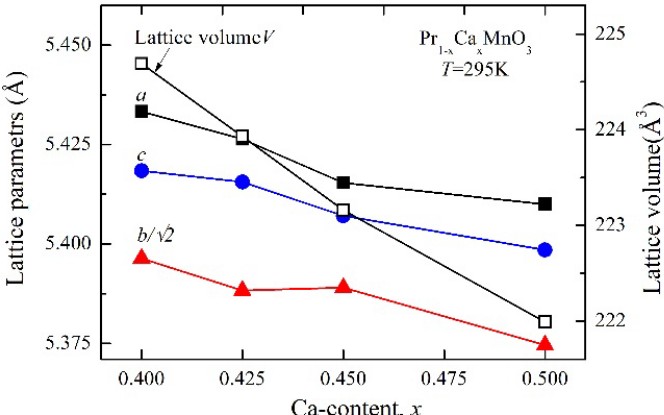

**Figure 1.** Room temperature variation of unit cell parameters (*a*, *b*, *c*) and unit cell volume, *V*, of $Pr_{1-x}Ca_xMnO_3$ with orthorhombic (*Pnma*) setting as a function of Ca content, *x*. Connecting lines exist to guide the eye.

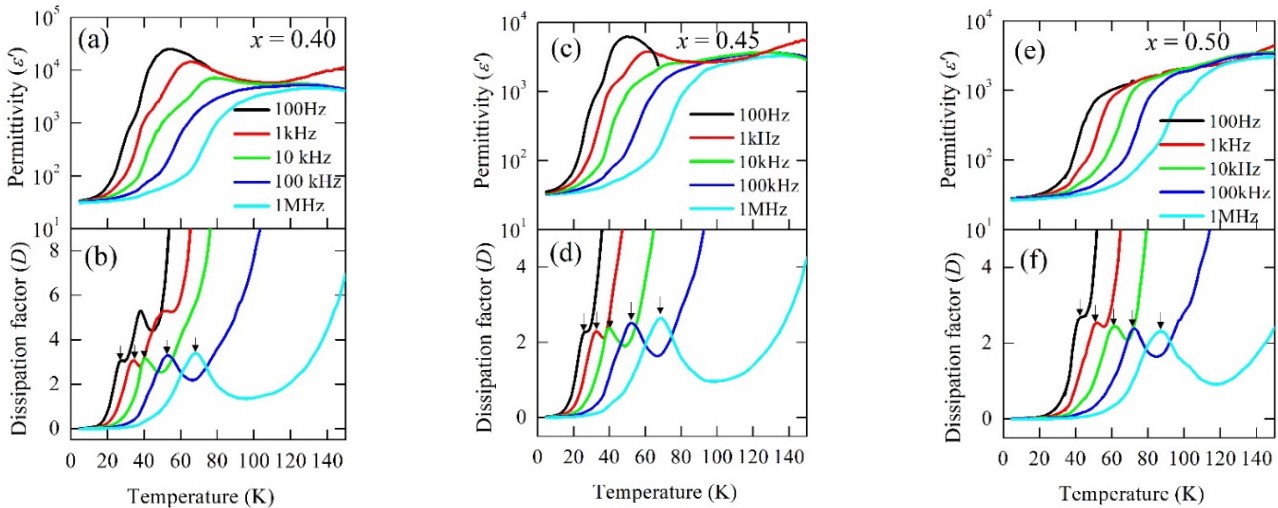

**Figure 2.** Dielectric properties of $Pr_{1-x}Ca_xMnO_3$ samples measured as a function of temperature for test frequencies 100 Hz, 1 kHz, 10 kHz, 100 kHz, and 1 MHz. The temperature variation of the real part of the dielectric permittivity ($\varepsilon'$) for Ca content (**a**) $x = 0.40$, (**c**) $x = 0.45$, and (**e**) $x = 0.50$ (upper panels) and dissipation factor (*D*) for (**b**) $x = 0.40$, (**d**) $x = 0.45$, and (**f**) $x = 0.50$ (lower panels). Arrows indicate the peak positions.

The isothermal measurements of the real part of the dielectric permittivity $\epsilon'(\omega)$ and dissipation factor $D(\omega)$ as a function of frequency were performed at $T = 4.5$ K and are shown in the upper and lower panels of Figure 3a, respectively. The dielectric response in PCMO at a higher temperature is usually dominated by the dielectric polarisation induced by polaron hopping and exhibits thermally activated relaxation behaviour [18].

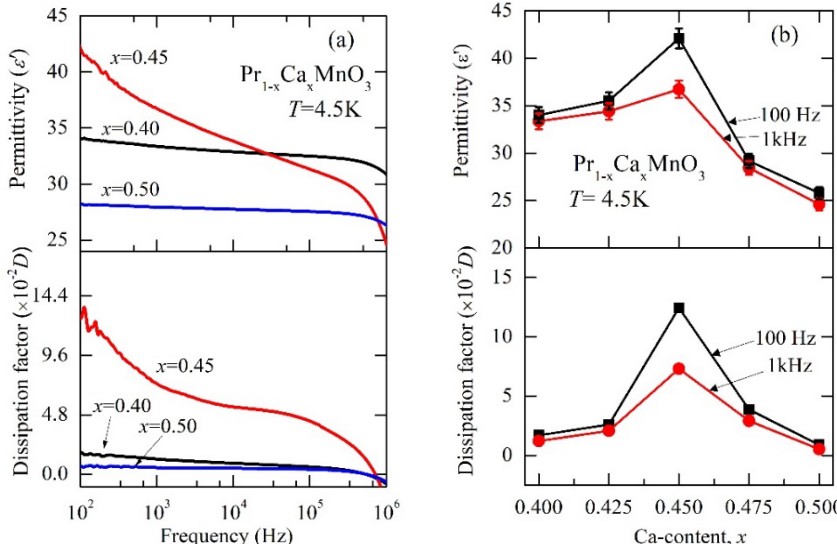

**Figure 3.** Dielectric properties of $Pr_{1-x}Ca_xMnO_3$ samples measured at 4.5 K. (**a**) Typical frequency dispersion pattern of the real part of dielectric permittivity (upper panel) and the dissipation factor (lower panel) for $Pr_{1-x}Ca_xMnO_3$ samples with different Ca content, $x$. (**b**) The Ca content, $x$, variations of the real part of the dielectric permittivity (upper panel) and dissipation factor (lower panel) for 100 Hz and 1 KHz as a function of $x$. Connecting lines exist to guide the eye.

However, at a sufficiently low temperature, the thermally activated charge carriers were frozen out, the dielectric response was free from the relaxation behaviour due to polaron hopping, and the dielectric response was dominated by the ionic lattice contributions [24,25]. As shown in Figure 3a, the frequency window (100 Hz to 1 MHz) over which the $\epsilon'(\omega)$ and $D(\omega)$ were measured was free from the typical dielectric relaxation due to polaron hopping and whatever frequency dispersion was observed at $T$ = 4.5 K arose from the ionic-lattice contributions. In Figure 3b, we plotted the variation of $\epsilon'(x)$ and $D(x)$ as functions of Ca content, $x$, for 100 Hz and 1 kHz frequencies. For both the frequencies, the $\epsilon'(\omega)$ and $D(\omega)$ exhibited similar $x$-dependence, and one can observe a conspicuous peak in $\epsilon'(x)$ and $D(x)$ at $x$ = 0.45.

The dielectric relaxation of PCMO originated from the polarisation induced by thermally activated polaron hopping and as a signature; the shifting of peak position, $T$, towards the higher temperature with increasing relaxation frequency was clearly observed. Due to the presence of strong electron-phonon interactions in the charge/orbital ordered manganite, the polaron formation was inevitable and polaron hopping conduction was expected. Generally, the variable range hopping model proposed by Mott [26] has been applied to the semiconductor-like conductivity of low-bandwidth manganates, such as PCMO at low temperature [17,18]. However, impurity hopping conductivity is expected at temperatures well below half of the Debye temperature, which is of the order of $\theta_{Debye}$~320 K [27]. In charge/orbital-ordered PCMO, below the $T_{CO}$, the charge careers became localised three-dimensionally due to by strong Coulomb interaction and also by the co-operative Jahn–Teller distortion of $MnO_6$ octahedra [10–12]. Therefore, hopping conduction in three dimensions is expected [17]. However, sometimes, the charge career localisation in the form of a two-dimensional stripe can also be observed for PCMO near $x$~0.7, i.e., above half filling [5]. Therefore, in order to understand the mechanism of dielectric relaxation due hopping conduction of a polaron, we analysed the temperature-dependent relaxation data for the temperature range of 25−90 K, using a small polaron hopping model (SPH) and also a three-dimensional Mott-variable-range hopping (VRH) model [17]. The relaxation frequency, $f$ (or relaxation time, $\tau$), of dielectric relaxation due to SPH is given by

$$fT = T\tau^{-1} = f_{01}\exp(-E_a/k_BT) \tag{1}$$

where $E_a$ is the activation energy of the polaron hopping conduction, $f_{01}$ is the pre-exponential factor, and $k_B$ is the Boltzmann constant. As shown in Figure 4a, the variation of $T\tau^{-1}$ was plotted as a function of $1000/T$ in the temperature range $25-90$ K. In the low-temperature side, certain deviations of the experimental data from the linearity were observed while the plotted data were fitted with SPH using Equation (1). From the linear fit, values of $E_a$ and $f_{01}$ were estimated and plotted as functions of $x$ in the upper and lower panels of Figure 5a, respectively.

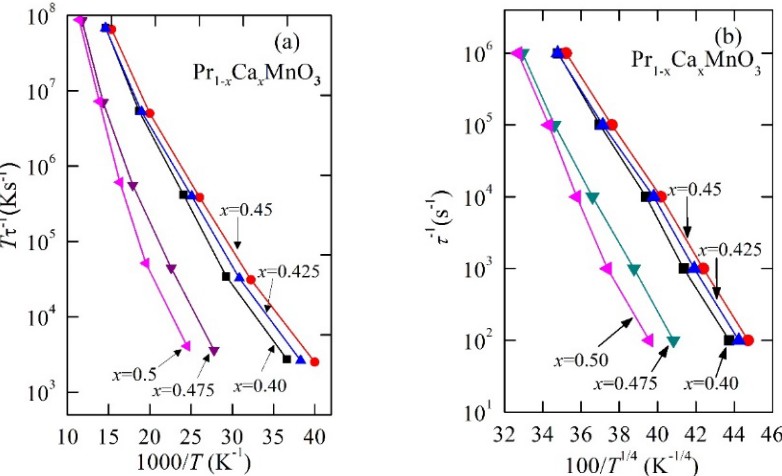

**Figure 4.** (**a**) The small polaron hopping (SPH) and (**b**) the variable range hopping (VRH) plots of dielectric relaxations of $Pr_{1-x}Ca_xMnO_3$ samples with different Ca content, $x = 0.40$, 0.425, 0.45, 0.475, and 0.50. Connecting lines exist to guide the eye.

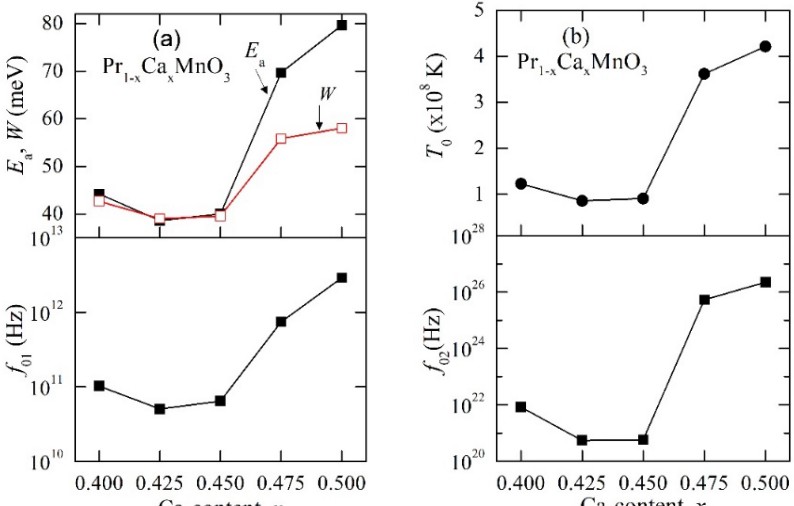

**Figure 5.** (**a**) Variation of activation energy, $E_a$ (upper panel), and pre-exponential factor, $f_{01}$ (lower panel), with $x$ of $Pr_{1-x}Ca_xMnO_3$ samples, obtained from the fitting of $T\tau^{-1}$ vs. $1000/T$ by the small polaron hopping model. (**b**) Ca content, $x$, dependence of $T_0$ and pre-exponential factor, $f_{02}$, estimated from the variable range hopping model. The Ca content $x$-dependence of the polaron hopping energy, $W$ (open symbol), estimated from $T_0$, is also plotted in the upper panel of (**a**) along with the $E_a$. Connecting lines exist to guide the eye.

It can be seen that both the parameters, $E_a$ and $f_{01}$, exhibited similar $x$ dependence. For $x = 0.40$ and 0.425, the $E_a$-value slightly decreased from 44.7 meV to 38.6 meV with an increase in $x$, while from $x = 0.45$ to 0.50, the $E_a$-value steadily increased from 40.3 meV to 79.6 meV with an increase in $x$. In an earlier publication, we reported similar $x$-dependence of the activation-energy-extracted temperature-dependent resistivity data of $Pr_{1-x}Ca_xMnO_3$ for $0.40 \leq x \leq 0.50$, wherein the minimum $E_a$-value was also found near

$x = 0.425$ [23]. Therefore, this particular Ca concentration, $x = 0.425$, roughly corresponded to a phase boundary between two different charge/orbital ordered phases. It is worth mentioning that the $E_a$-values estimated from the dielectric relaxation were fairly similar to the activation energy for polaron hopping estimated from the resistivity data, which strongly indicates the polaronic nature of charge-ordered PCMO samples [17,18].

The Mott VRH model [17,26], based on the thermally assisted hopping of charge carriers between the localised states near the Fermi level, is given by Equation (2), where $f_{02}$ is the pre-exponential factor; $T_0$ is a characteristic Mott temperature that is given by $T_0 = 24/\pi k_B N(E_F)\xi^3$, where $N(E_F)$ is the density of localised states at the Fermi level and $\xi$ is the decay length of the localised wave function; and $\beta = \frac{1}{d+1}$ for $d$-dimensional conduction.

$$f = \tau^{-1} = f_{02} \exp\left(-(T_0/T)^\beta\right) \tag{2}$$

In Figure 4b, we plotted the variation of $\tau^{-1}$ as a function of $100/T^{1/4}$ in the temperature range 25–90 K, i.e., according to the three-dimensional VRH model. The linearity of the plot indicates that the conduction mechanism in PCMO was governed by thermally assisted hopping of charge carriers in three dimensions. It is important to note that the SPH plots showed certain deviations from the linearity, which indicates that VRH was the dominant mechanism for charge conduction of PCMO in the measured temperature range. Mott activation energy, $T_0$, and pre-exponential factor, $f_{02}$, were estimated from the linear fitting $\tau^{-1}$ vs. $100/T^{1/4}$ plot and plotted as a function of $x$ in the upper and lower panels of Figure 5b, respectively. Similar to the SPH model, both the parameters exhibited a broad minima near $x = 0.425$. In the Mott VRH mechanism, the hopping energy, $W$, of a polaron can be estimated using the relation $W = 0.25 k_B T_0^{1/4} T^{3/4}$. Taking $T = 50$ K, an average temperature around which the dielectric relaxation was analysed, we obtained the value of polaron hopping energy, $W$, for PCMO samples and plotted them as a function of $x$ in the upper panel of Figure 5a alongside of $E_a(x)$ for comparison. The $W(x)$ also showed a similar Ca content dependence to that of the activation energy, $E_a(x)$, estimated from the SPH model.

## 3. Discussion

Let us turn our discussion towards the possible origin of observed anomalous $x$-dependence of dielectric properties. Presently, the origin observed peak in the $x$-dependence of the dielectric permittivity of PCMO near $x{\sim}0.45$ is not clearly understood. However, two possible explanations can be drawn on the basis of earlier literature: One possibility is related to the inhomogeneous CB-type charge-ordered phase. The CB-type charge/orbital ordered phase in PCMO for $x \leq 0.50$ was susceptible to phase separation due to the presence of an unequal amount of $Mn^{3+}$ and $Mn^{4+}$ ions. The evolution from homogeneous to inhomogeneous CB-type charge-ordered phase with decreasing $x$ was found at a critical doping near $x = 0.425$ [22,23]. Therefore, the observed enhancement of a dielectric constant near $x = 0.45$ may roughly correspond to the phase boundary between inhomogeneous and homogeneous phases.

Another possibility is based on the formation of a ZP-type ordered phase. In this context, using the tight binding model calculation based on the degenerate double exchange framework, Efremov et al. showed that the character of the charge/orbital-ordered phase in PCMO was not essentially the same over the doping range $0.40 \leq x \leq 0.50$; instead, the character of the ordered phase changed in a systematic manner from $x = 0.50$ down to $x = 0.40$ [16]. The charge/orbital ordering pattern was found to have evolved from a pure CB-type charge-ordered phase at $x = 0.50$ to an admixture phase of CB-type- and ZP-type-ordered phases with decreasing $x$ and finally to a pure ZP-type ordered phase at $x = 0.40$. The admixture phase can break the inversion symmetry and leads to the development of a local ferroelectric moment. In fact, the direct experimental evidence of subtle macroscopic ferroelectric polarisation has recently been found in remnant electrical polarisation measurements in PCMO samples [28,29], while indirect evidence of a local

ferroelectric moment was confirmed earlier in EFG measurement [22]. Therefore, the observed peak in $\epsilon'(x)$ near $x\sim0.45$ may have a trace of the local ferroelectric moment due to the presence of the admixture phase composed of CB-type- and ZP-type-ordered phases. However, direct measurement of the macroscopic electric polarisation is difficult because of relatively high conductivity of PCMO samples (dissipation factor $\approx0.03$), which can obscure the finite macroscopic polarisation if it exists, at least at the local scale. In addition, the charge-ordered state in PCMO can be collapsed by a sufficient electric field and it provides a huge electroresistance effect that further complicates the experimental situation in terms of directly probing the macroscopic polarisation [30]. Furthermore, the PCMO sample with $x = 0.45$ exhibited a stronger frequency dispersion in $\epsilon'(\omega)$ and $D(\omega)$ than the two end compositions $x = 0.40$ and 0.50. Typically, the amount of frequency dispersion in $\epsilon'(\omega)$ and $D(\omega)$ depends on the degree of phase separation (electronic inhomogeneity) present in a system. Therefore, the larger amount of frequency dispersion observed in $\epsilon'(\omega)$ and $D(\omega)$ for the sample with $x = 0.45$ indicated that this intermediate composition possesses a stronger tendency towards phase separation than the two end compositions due to coexistence of ZP-ordered and CB-type-charge/orbital-ordered phases [31]. Therefore, the observed result strongly suggests the presence of an admixture phase in the intermediate composition range.

## 4. Materials and Methods

For this study, polycrystalline samples with nominal compositions of $Pr_{1-x}Ca_xMnO_3$ ($x = 0.40, 0.425, 0.450, 0.475$, and 0.50) were synthesised by a conventional solid state reaction method. The stoichiometric mixtures of the starting materials, $Pr_2O_3$ (3N), $CaCO_3$ (3N), and $MnO_2$ (3N), were calcined at 1373 K in air for 24 h. The calcined powders were pressed into circular pallet form and sintered at 1673 K in air for 48 h. The detailed structural characterisations and physical property measurements were carried out on the as-synthesised samples that were reported in our earlier publication [28]. The temperature-dependent dielectric property was measured in the frequency range of 100 Hz to 1 MHz by using an LCR meter (Kesight) from 4 to 150 K in closed cycle cryostat. The circular disc-shaped samples with a typical diameter of $\approx8$ mm and thickness of $\approx0.5$ mm were used for the measurement. For the electrode, air-drying silver pasted was coated on pallet surfaces and cured at 623 K.

## 5. Conclusions

In summary, the low-temperature dielectric properties of charge-ordered manganite $Pr_{1-x}Ca_xMnO_3$ for $0.40 \leq x \leq 0.50$ were investigated systematically as a function of Ca content, $x$. The $x$ dependence of dielectric permittivity exhibited distinct maxima near $x = 0.45$. The overall dielectric response exhibited thermally activated relaxation behaviour due to polaron hopping conduction and the estimated polaron transport parameters, i.e., the activation energy of polaron hopping and polaron hopping frequency also showed broad minima between $x = 0.425$ and 0.45. The enhancement of dielectric constant near $x = 0.45$ may roughly correspond to the phase boundary between inhomogeneous and homogeneous phases. On the other hand, if we assume the ZP-type order phase below the boundary, $x = 0.425$, of course, another possibility will be the phase boundary between CB-type charge/orbital phase and ZP-type-ordered phases, and the observed peak in $\epsilon'(x)$ near $x\sim0.45$ may provide a trace of the local ferroelectric moment due to the presence of admixture phase composed of CB-type-charge/orbital-ordered and ZP-type-ordered phases.

**Author Contributions:** Supervision, conceptualisation, reviewing and editing: P.S.M.; methodology, validation: P.S.M. and A.K.S.; formal analysis, original draft preparation, visualisation: A.K.S. All authors have read and agreed to the published version of the manuscript.

**Funding:** This research was partially funded by DST-SERB, Government of India (grant no. CRG/2019/00006618).

**Data Availability Statement:** Not applicable.

**Acknowledgments:** The authors also gratefully acknowledge Ichiro Terasaki and Ryuji Okazaki for valuable discussions.

**Conflicts of Interest:** The authors declare no conflict of interest.

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
