# Peer review of "Anomalous Ca Content Dependence of Dielectric Properties of Charge-Ordered Pr1−xCaxMnO3 as a Signature of Charge-Ordered Phase Modulation"

_condensedmatter, doi:10.3390/condmat7040058_

Round 1

Reviewer 1 Report

The authors studied the Ca-content dependence of low-temperature dielectric property in Pr1-xCaxMnO3 (0.40 ≤x≤ 0.50), focusing on the charge/orbital ordered state. They observed the Ca-content dependence of dielectric permittivity and dissipation factor exhibit distinct maxima around x=0.45. They further analyzed their results by SPH and VRH models and found that the polaron parameters display non-monotonic variation with x and exhibit broad minima between x=0.425-0.45. They interpret this anomaly at x=0.425-0.45 as a possible modulation of the checkerboard type charge ordering pattern in this system.

The present manuscript is relevant for the field of charge/orbital ordered manganite. The paper is well written and well organized.
Their results are important and provide new knowledge to the field. However, there are a few questions needed to be addressed before I could recommend the manuscript be published in Condensed Matter.

1. The horizontal axis label in Fig 1 is not defined. It would be better to add 'Ca-content, x'.
2. In Eq.1, T_p \tao^(-1) is defined. However, in Line 133, the authors used T \tao^(-1). In Fig4, T \tao^(-1) is used.
   Could the authors keep them in the same style?
3. The SPH and VRH model are not properly referenced in Line 128-Line130 and Line152.
4. The physical meaning of T_0 in Line 154 is not properly introduced and explained.
5. Line157, it is not clear to me why VRH model works better than the SPH model. The authors could explain more here.

Minor:
Line 20:checker board--> checkerboard
Line121: dispersion less--> dispersionless
Line 124:  observed--> observe
Line 133: Boltzmann’s constant.-->Boltzmann constant.

Author Response

Kindly find response in attached file

Reviewer 2 Report

The manuscript titled “Anomalous Ca-content dependence of dielectric property of charge ordered Pr1-xCaxMnO3 as a signature of charge-ordered phase modulation” written by Ankit Kumar Singh et al. introduced how the Ca-content modulates the phase of Pr1-xCaxMnO3 and changes its dielectric properties. Through the description of experimental results, the author obtained the specific value or interval corresponding to the Ca-content when the dielectric properties of Pr1-xCaxMnO3 reach the minimum or maximum value, respectively. In addition, the dielectric relaxation behaviour over the investigated temperature range was analysied with the help of small polaron hopping model and as well as variable range hopping model. This work is comprehensive, valuable and logical, which means the manuscript is worth publishing if the following issues are addressed.

1)   At the part of “Introduction”, the author gave a number of examples of Ca-content affecting the dielectric properties of charge ordered material Pr1-xCaxMnO3 through phase modulation. However, all of the demonstrations were listed in a situation where is slightly lack of logic and organization. It will be better if all kinds of examples can be summarized and stated in a logical order.

2) In the description about Fig. 2 and elsewhere, the relationship between temperature and permittivity or dissipation factor at various frequency and Ca-content were showed in detail. Advice to the author to add more theoretical analysis or introduction about “charge-ordered phase modulation”, “small polaron hopping model” and “variable range hopping model” so that the experimental results from the Fig. 2 ect. are more convinced.

3) In the whole article, there are some repetitive descriptions about the effect on Pr1-xCaxMnO3’s phase for Ca-content , resulting in a little confusion in the structure of the article. It is helpful to properly organize repetitive descriptions of articles and highlight key points.

Author Response

Kindly find the response in attached file

Reviewer 3 Report

The authors investigate, in the temperature range below 140 K, two dielectric properties [dielectric permittivity and dissipation factor] of Pr1-xCaxMnO3 over the doping range 0.40 ≤ x ≤ 0.5, and observe anomalous x-dependent behavior of these properties around x=0.45. They use two theoretically models, i.e., small polaron hopping and variable range hopping. As a possible origin of the anomaly in dielectric properties they endorse the modulation/evolution of the charge-ordered phase.

In my opinion, this work is not mature for publication. I will give some reasons below:

Α. Bad syntax and grammar starting even from title and abstract. Some examples:

1. "Anomalous Ca-content dependence of dielectric property of charge ordered Pr1-xCaxMnO3 as a signature of charge-ordered phase modulation"

could be

"Anomalous Ca-content dependence of the dielectric properties of Pr1-xCaxMnO3 as a signature of charge-ordered phase modulation"

2. "Low temperature dielectric property of charge/orbital ordered manganite, Pr1-xCaxMnO3 for 0.40 ≤ x ≤ 0.50 is investigated systematically as a function of Ca-content, x."

should be

"The low-temperature dielectric properties of charge/orbital ordered manganite, Pr1-xCaxMnO3, for 0.40 ≤ x ≤ 0.50, is investigated systematically as a function of the Ca molecular fraction, x."

3. "The dielectric relaxation behaviour over the investigated temperature range is analysed with the help of small polaron hopping model and as well as variable range hopping model. "

should be

"The dielectric relaxation behaviour over the investigated temperature range is analysed with the help of two models: the small polaron hopping model and the variable range hopping model."

4. "The physical property R1−xAxMnO3 involves complex interplay among charge, spin and orbital degree of freedoms."

should be

"The physical properties of R1−xAxMnO3 involve complex interplay among charge, spin and orbital degree of freedoms. "

5. "and etc" should be "and others "

and many many more.

Β. In the manuscript there is no information about sample preparation, dimensions and geometry, which is important not only experimentally but also for possible explanation using a small polaron hopping or a variable range hopping model. Please see relevant comment below.

C. Is the sentence "Often the competitive phase in manganite may co-exist over a range of length scales as a signature of electronic phase separation." relevant to this manuscipt? Manganite is monoclinic manganese oxide-hydroxide, MnO(OH). Is there presence of oxygen or hydrogen in these samples? Is the word manganite used in a broader sense in this manuscript? If yes, the readers should be informed for this broader use.

D. There are various small polaron hopping models. A general law for small polaron hopping, based on percolation arguments, is the T^{−ε/(ε+r)} law, where, ε is the number of energies involved in the percolation condition and r is the number of the spatial dimensions involved. For the low temperature bulk case (ε = 1 and r = 3) the T^{−1/4} law (Mott’s law) is obtained. For the case of longitudinal conduction at low temperatures in thin films (ε = 1 and r = 2), a T^{−1/3} law is obtained. For 1D case at high temperatures (ε = 2 and r = 1), a T^{−2/3} law is expected, while at low temperatures (ε = 1 and r = 1) a T^{−1/2} law, in accordance with the variable range hopping results. The authors refer to small polaron hopping and to variable range hopping without any references, which is unacceptable.

E. The density of states and extent of wave function are also two crucial factors for

small polaron hopping conductivity in 1D. References here are also necessary, I guess.

F. For the piece of text:

"For x = 0.40 and 0.425, the Ea-value slightly decreases from 44.7 meV to 38.6 meV with increase of x while for x = 0.45 to 0.50 the Ea-value steadily increases from 40.3 meV to 79.6 meV with increase of x."

I would like to notice that this is probably a result of taking a particular small polaron model [not cited!] and ignoring the number of energies involved in the percolation condition, the number of the spatial dimensions involved as well as the density of states and extent of wave function, please see above.

G. How was the dielectric permittivity (and the dissipation factor) vs. frequency measured? There is no such information in the manuscript, I guess.

H. Apart from those points, this work seems a continuation of reference [28]. The authors should explain better the difference of the two works.

Author Response

Kindly find the response letter in attached file.

Round 2

Reviewer 1 Report

The authors have improved the draft following my suggestions.

Reviewer 3 Report

The authors have, more or less, satisfactorily replied to my remarks and comments. The article could be still improved, especially by citing appropriate work on small polarons, but I have no objection to publish it as is.